# Lignin/Carbohydrate Complex Isolated from *Posidonia oceanica* Sea Balls (Egagropili): Characterization and Antioxidant Reinforcement of Protein-Based Films

**DOI:** 10.3390/ijms22179147

**Published:** 2021-08-24

**Authors:** Seyedeh Fatemeh Mirpoor, Odile Francesca Restaino, Chiara Schiraldi, Concetta Valeria L. Giosafatto, Francesco Ruffo, Raffaele Porta

**Affiliations:** 1Department of Chemical Sciences, University of Naples “Federico II”, Montesantangelo Campus, via Cintia 4, 80126 Naples, Italy; seyedehfatemeh.mirpoor@unina.it (S.F.M.); giosafat@unina.it (C.V.L.G.); Francesco.ruffo@unina.it (F.R.); 2Department of Experimental Medicine, Section of Biotechnology and Molecular Biology, University of Campania “Luigi Vanvitelli”, 80138 Naples, Italy; odilefrancesca.restaino@unicampania.it (O.F.R.); chiara.schiraldi@unicampania.it (C.S.)

**Keywords:** lignin, egagropili, *Posidonia oceanica*, protein-based films, antioxidant

## Abstract

A lignin fraction (LF) was extracted from the sea balls of *Posidonia oceanica* (egagropili) and extensively dialyzed and characterized by FT-IR and NMR analyses. LF resulted water soluble and exhibited a brownish-to-black color with the highest absorbance in the range of 250–400 nm, attributed to the chromophore functional groups present in the phenylpropane-based polymer. LF high-performance size exclusion chromatography analysis showed a highly represented (98.77%) species of 34.75 kDa molecular weight with a polydispersity index of 1.10 and an intrinsic viscosity of 0.15. Quantitative analysis of carbohydrates indicated that they represented 28.3% of the dry weight of the untreated egagropili fibers and 72.5% of that of LF. In particular, eight different monosaccharides were detected (fucose, arabinose, rhamnose, galactose, glucose, xylose, glucosamine and glucuronic acid), glucuronic acid (46.6%) and rhamnose (29.6%) being the most present monosaccharides in the LF. Almost all the phenol content of LF (113.85 ± 5.87 mg gallic acid eq/g of extract) was water soluble, whereas around 22% of it consisted of flavonoids and only 10% of the flavonoids consisted of anthocyanins. Therefore, LF isolated from egagropili lignocellulosic material could be defined as a water-soluble lignin/carbohydrate complex (LCC) formed by a phenol polymeric chain covalently bound to hemicellulose fragments. LCC exhibited a remarkable antioxidant activity that remained quite stable during 6 months and could be easily incorporated into a protein-based film and released from the latter overtime. These findings suggest egagropili LCC as a suitable candidate as an antioxidant additive for the reinforcement of packaging of foods with high susceptibility to be deteriorated in aerobic conditions.

## 1. Introduction

Lignocellulosic biomass derived from woody and non-woody dry matter, such as grasses, trees, as well as harvest residues from food crops, is the largest amount of sustainable carbon-containing resources on earth that are annually produced (around 181.5 billion tons) [1,2,3,4,5,6]. Lignocellulosic material contained in plant cell walls is mainly constituted by three kinds of polymers, i.e., cellulose, hemicellulose and lignin, the content of which is affected by various factors, such as the different species, source and type (hardwood, softwood or grass) of the original biomass [7,8]. Cellulose is a homogeneous and linear polysaccharide consisting of repeating units of cellobiose monomer, β-1,4-linked anhydro-D-glucose, tightly packed in the crystalline parts of cellulose fibrils via intra- and inter-molecular hydrogen bonds [9]. Hemicellulose is an amorphous, short chain and heterogeneously branched polymer constituted by both pentoses and hexoses, mainly xylans (arabinoxylans and 4-O-methyl-glucuronoxylans), galactomannans, glucomannans and xyloglucans (4-linked β-D-glucans with attached side chains) [10]. The dominant polymers occurring in hemicellulose extracted from hardwood are xylans, while those present in softwood are glucomannans [11,12]. Hemicellulose and cellulose polymers are linked by hydrogen bridges and Van der Waal’s interactions, while hemicellulose adheres to the lignin through phenyl glycoside as well as through esters and benzyl ether covalent bonds, giving rise to lignin-carbohydrate complexes (LCCs) [13,14]. Extracted hemicellulose easily degrades to its constituent sugar units, mainly xylose (Xyl), mannose, arabinose (Arab), glucose (Glc), glucuronic acid (GlcA) under acid hydrolysis [15].

Lignin, a highly branched and amorphous biomacromolecule, is the second most abundant aromatic biopolymer in nature [16]. It consists of three different phenylpropane monomers known as p-hydroxyphenyl, coniferyl and sinapyl alcohols, containing zero, one, or two methoxyl groups, respectively [17,18]. The constituent structure and the amount of lignin occurring in plant cell wall depend on the origin, amount of different mono-lignols, chemical bonds in the polymer structure and on the source of the lignocellulosic biomass [19,20]. Coniferyl alcohol is the dominant structure occurring in the softwood lignin, while in the hardwood lignin commonly exist both coniferyl and sinapyl alcohols, the second one being dominant [21]. Lignin monomers are linked by several types of carbon–oxygen (aryl-ether) and carbon–carbon interactions, 50% of which are β-O-4′ ether linkages, whereas the others are β-5 phenylcoumaran, β-β′ resinol, α-O-4′ ether, 4-O-5′ diphenyl ether, 5-5 biphenyl and β-1′ diphenyl methane bonds [22,23,24]. The monomer random distribution gives rise to heterogeneous lignin structures difficult to isolate, standardize and characterize [25].

Lignin plays a key role in protecting plants against possible environmental stresses by providing a marked mechanical support to their cell walls [26]. Due to the aromatic structure and multiple functional groups, lignin possesses also several physicochemical and biological properties, such as fire resistance, wettability, biodegradability, as well as antioxidant, antifungal and antimicrobial activities [27,28,29,30,31]. All these features might be affected by the method of extraction of the different lignin polymers present in the different lignocellulosic sources [1,2,24,32], since, during the isolation process, lignin fragments into products with different lower molecular weight (Mw) [1,2]. Thus, temperature, pH and pressure of the system, as well as the capacity of the solvent to dissolve the extracted polymer fragments, may significantly affect their physicochemical and biological properties [24,33,34]. However, lignin extraction from lignocellulosic biomass is generally performed by gradually breaking down the polymer into lower Mw products [1,2]. Around 85% of the produced lignin is obtained through the “kraft pulping process”, able to break down the majority of hemicellulose covalently bound to lignin and to give rise to LCCs that are different depending on the specific lignocellulosic biomass used [35,36] (Figure 1A).

In the present study, LCCs were extracted from egagropili (fibrous balls from *Posidonia oceanica* detritus), considered as a waste since they accumulate in huge numbers along sandy shores, by using a sodium chlorite extraction procedure at acidic pH [37]. With this procedure, as shown in Figure 1B, coniferyl aldehyde and aromatic ketones present in the lignin originally linked to hemicellulose undergo to oxidative and ring-opening reactions to form acidic groups, making lignin fragments more soluble in water [38]. The isolated LCCs were then characterized and used as reinforcement of protein-based films with the aim to replace oil-based polymers in packaging systems.

## 2. Results and Discussion

### 2.1. Egagropili Powder Fractionation and Fourier-Transform Infrared Spectroscopy Analysis

In order to separate lignin, holocellulose and cellulose fractions, egagropili powder was preliminarily washed, grinded, dewaxed and heat-dried. Fourier-transform infrared (FT-IR) spectroscopy was utilized to determine the chemical structure of egagropili fibers, as well as that of both holocellulose and cellulose fractions [39]. FT-IR spectra reported in Figure 2 show the presence of lignin signals only in the spectrum corresponding to the untreated egagropili fibers (Figure 2A: 1457 cm^−1^ attributed to the CH_2_ and CH_3_ of C-H bending of lignin; 1507 cm^−1^, 1595 cm^−1^ and 1630 cm^−1^ ascribed to the in plane stretching vibration of the C=C and C=O of lignin aromatic ring) [39,40,41].

In addition, the C=O ester peak signaled at 1735 cm^−1^ represents the stretching vibration of the carbonyl and acetyl groups occurring in the Xyl probably present in both lignin and unconjugated hemicellulose [41,42,43,44]. It is worthy to note that none of the abovementioned peaks were present in the obtained cellulose spectrum (Figure 2C). Furthermore, the absorption band observed around 1032 cm^−1^ in the holocellulose spectrum (Figure 2B), but absent in the cellulose spectrum (Figure 2C), is probably due to the C-O stretching in pyranose ring that is an indicator of the presence of hemicellulose [43,44], whereas the peak at around 1247 cm^−1^, present in the spectrum shown in Figure 2C, confirms the removal of hemicellulose from the cellulose fraction. Finally, the peaks at around 898 cm^−1^ and 1105 cm^−1^, associated with the characteristics of the β-(1→4)glycosidic bond and C-O-C glycosidic ether bond, respectively, are strictly related to cellulose [44,45]. Therefore, cellulose typical peaks appeared in all three spectra, whereas their intensity increased in each step of the chemical bleaching treatment.

### 2.2. Egagropili Lignin Fraction Characterization

Lignin represents a class of complex aromatic polymers built up of phenylpropane units and is one of the most abundant organic materials and renewable resources in Nature. Its composition varies among species, phylogenetic groups, cell types and developmental stages. Lignin is a by-product of several industries and bio-refinery processes and may have different structures and functional groups, as well as different physicochemical and biological properties, according to the extraction procedure that it undergoes. In fact, the native structure of lignin degrades during its extraction from the original lignocellulosic material and smaller polymer molecules endowed with new functional groups are formed. Lignin fraction (LF) obtained from egagropili by sodium chlorite oxidation technique [37] was water soluble and exhibited a brownish-to-black color, probably due to the presence of unsaturated functional groups, including conjugated carbonyl groups, aromatic rings and carbon–carbon double bonds, produced during such harsh chemical treatment [46,47,48]. As shown in Figure 3, the highest absorbance of the extensively dialyzed LF is in the range of 250–400 nm that could be attributed to the different chromophore functional groups present in the extracted phenylpropane-based polymer. Similar results have been reported by Rukmanikrishnan et al. [49] and Lee et al. [50], who used lignin as a UV blocker component in biopolymers, such as gellan gum or 2-hydroxyethyl cellulose, and as an active ingredient for sunscreens.

A proton NMR spectrum of egagropili LF was also recorded in D_2_O at pH 12, and the relative spectrum, reported in Figure 4, shows that the prevailing signals are broad peaks in the aromatic (6.5–7.5 ppm) and alkoxide (3.5–4.5 ppm) regions. The latter signal, which is due to the -OCH_3_ groups on the syringyl and guaiacyl units, appeared relatively intense. This observation, indicative of the presence of additional protons of alkoxy nature, suggested checking for the possible existence of sugar residues in this fraction.

Therefore, to investigate whether hemicellulose-deriving carbohydrate fragments remained linked, either through phenyl glycoside, esters or benzyl ether covalent bonds, to the phenolic polymer during lignin extraction from the egagropili lignocellulosic material, monosaccharide composition analyses were carried out on the egagropili-derived fibers, LF, hollocellulose and cellulose. Figure 5 shows that eight different monosaccharides were detected in the untreated egagropili fiber sample: neutral sugars like fucose (Fuc), Arab, rhamnose (Rham), galactose (Gal), Glc and Xyl were identified, as well as an amino-sugar, glucosamine (GlcN), and an uronic acid such as GlcA (Figure 5A–D). Their representativity ranged from 51.0% of Glc to the 0.6% of Fuc (panels A and D). The same monosaccharide composition was also found in the LF (panels B and D) but with different percentages, as Glc presence was greatly reduced. Conversely, only Glc was detected in the hollocellulose and cellulose samples (panels C and D). Quantitative analyses indicated that carbohydrates were 28.3% of the dry weight of the untreated egagropili fibers and 72.5% of the LF dry weight. Fuc, Arab, Rham, Gal, Glc, Xyl, and GlcA were previously found also in the water extracts of *Posidonia oceanica* leaves, together with other sugars like mannose and galacturonic acid, even though their relative abundance was quite different with percentages that varied up to 80% for Gal and up to 50% and 45% for Glc and Xyl, respectively [51]. Therefore, we conclude that LF obtained from egagropili contained one or more LCC(s) (27.5 lignin:72.5 carbohydrate).

Furthermore, total phenol content (TPC) as well as flavonoid and anthocyanin content of egagropili LF were investigated. It is well known that phenolic compounds are the largest group of phytochemicals that are responsible for antioxidant activity in plants and in their derived products. They possess an aromatic ring with one or more hydroxyl groups with a wide range of biological properties, such as anti-mutagenic, antioxidant and anti-carcinogenic activities [52,53]. The TPC value determined for egagropili LF (113.85 ± 5.87 mg gallic acid eqs/g of extract), reported in Table 1, was meaningful, being in the same range of that (92.7 to 181.6 mg gallic acid eqs/g) measured by Faustino et al. [54] in the industrial black liquors obtained by two cooking processes (kraft and sulphite) to produce *Eucalyptus globulus* pulp. Moreover, it is worthy to note that almost all the phenol content occurring in the LF was water soluble since only a small amount of it was extracted by hexane (Table 1). However, as it was previously reported [55], lignin TPC is affected by several factors such as the plant age and harvesting time, the specific method of lignin extraction, as well as the storage period of the extracted lignin.

The contribution of both flavonoids and anthocyanins in the egagropili LF TPC was also evaluated. Flavonoids represent the main group of phenolic compounds present in plants either as glycosides or in free state, being constituted by two benzene rings separated by a propane unit. They possess several biological activities, such as antiulcer, anti-arthritic, anti-angiogenic, anticancer and antimicrobial, as well as antioxidant properties due to their polyphenolic nature, which enable them to scavenge free radicals [56]. The data reported in Table 1 indicate that almost 22% of the TPC of egagropili LF is constituted by flavonoids and that, in particular, only 10% of them are anthocyanins, water-soluble but unstable pigments, because they easily decompose during the extraction, purification, and storage processes [57,58].

Finally, in order to determine the Mw of egagropili LF component(s), a size-exclusion chromatographic analysis with triple detector array system (SEC-TDA) was carried out. The SEC-TDA molecular separation showed a particularly homogeneous sample containing a highly represented (98.77%) species of 34.75 kDa with a polydispersity index of 1.10 and an intrinsic viscosity of 0.15 (Figure 6A,B). Although there are numerous studies reported in literature on the Mw of lignin originating from different biomass sources, this is the first time to our knowledge that the Mw of a LF that contains one or more LCC, from egagropili, has been determined; as well, it is also the first Mw analysis in which a SEC-TDA method has been applied to lignin polymer(s). Lignin Mw values reported so far range from 5.9 to 74.8 kDa, and it is known that they could change according to the lignin extractive methods used, such as milled wood process, enzymatic hydrolysis and/or mild acidolysis, as well as according to the origin of the biomass (e.g., from soft or hard wood) and the different instrumental techniques [59]. Gel permeation chromatography (GPC) and SEC have been frequently employed to determine the Mw values of different types of lignin polymers of different origins [59]. In particular, GPC coupled with a detector based on low-angle laser light scattering resulted more precise and accurate in determining lignin Mw than GCP-MALDI-TOFMS methods [59]. However, no data have been reported so far on the use of SEC-TDA as a tool to determine the Mw of lignin-containing fractions. In particular, with regard to the egagropili, literature data have reported so far only the analysis of the Mw of a “milled-wood” LF″ extracted from *P. oceanica* sea balls by using a classic gel-permeation chromatography method. Compared to the Mw data obtained for LF described in the present study, this “milled-wood” lignin sample showed lower Mw (6.1 kDa) but higher polydispersity (Mw/Mn = 2.2) [60]. That difference is probably due to the different extractive method used leading to a LF containing only lignin and not to a water-soluble lignin/carbohydrate complex [60].

### 2.3. Egagropili Lignin Fraction Antioxidant Activity

The significant phenol content observed in the egagropili LF suggested to investigate the antioxidant activity by hydrogen donation and single-electron transfer reactions [55]. As shown in Figure 7, antioxidant activity of LF was found to linearly increase by increasing its concentration from 0.03 to 0.15 mg up to 15 min, then the curves reached the plateau indicating 45 and 75% of the 2,2-diphenyl-1-picrylhydrazyl (DPPH) scavenging activities at 90 min of sample incubation by using the lowest and the highest LF amount, respectively. It is noteworthy that even a very small amount of egagropili LF exhibited a remarkable antioxidant activity similar to that of lignin obtained from spruce and pine [61].

Furthermore, the effect of LF storage on the antioxidant activity was studied over time. Figure 8 reveals that LF antioxidant activity remained quite stable during 6 months since the DPPH scavenging activity slightly changed, under the experimental conditions selected, only from 83 to 79%. These findings strongly suggest the potential exploitation of egagropili LF as a stable and prominent antioxidant agent. Similar results were obtained by Alzagameem et al. [55] and Dizhbite et al. [61], who reported that DPPH sca-venging activity of different fractions of Kraft lignin from different sources slightly decreased after their storing for 6 months. However, the antioxidant activities exhibited by all the previously extracted LFs resulted lower in comparison with that of LF isolated in the present study from egagropili by oxidation with sodium chlorite.

### 2.4. Antioxidant Activity of Hemp-Protein-Based Films Containing Egagropili Lignin Fraction

The effect of different concentrations of egagropili LF on the mechanical and barrier properties of hemp-protein (HP)-based films has been previously investigated [37]. Since it was demonstrated that the presence of 6% LF (*w*/*w* protein) in the film-forming solution (FFS) gave rise to films with improved performances, the antioxidant activity of both FFSs and derived films was studied by measuring their DPPH radical scavenging activity in the presence of the same amount of egagropili LF (Figure 9).

The obtained results indicate that both FFSs and the derived films prepared with HPs in the absence of LF exhibited a very low antioxidant activity measured at different times of incubation with DPPH solution. Conversely, when both HP-based FFSs and films were obtained in the presence of 6% egagropili LF, their DPPH scavenging activities strongly increased (about 10 times). It is worthy to note that the antioxidant activity of the films, as well as that of their corresponding FFSs, increased as a function of the increasing incubation times of the samples. Similar results were obtained by Mohammad Zadeh et al. [62] with soy protein isolate-based films added with lignosulfonate lignin.

Finally, Figure 10 shows the release of antioxidant compounds from the matrix of HP-based films containing 6% LF into the DPPH solution during 24 h incubation at both dark and room temperature. Of note, 30% of the antioxidant activity seems to have been released from the films after 2 h, whereas 50% of the DPPH scavenging activity was measured only after 22 h from the moment in which the films were immersed into the DPPH solution.

As is shown in Figure 10, the control films prepared in the absence of egagropili LF were not able to release into DPPH solution any antioxidant agent, since no DPPH scavenging activity was detected at all over time. Therefore, taking also into account the increased barrier effect of LF-containing protein-based films [37], egagropili LF might be considered a suitable candidate as an antioxidant additive for reinforced packaging of food products with high susceptibility to be deteriorated in aerobic conditions.

## 3. Materials and Methods

### 3.1. Materials

Hemp oilseed cakes, a generous gift of prof. Daniele Naviglio, were purchased from Consorzio Goji Italia (Andria, Italy). Egagropili sea balls were collected in the sardinian Poetto beach (Cagliari, Italy) and stored at 4 °C until used. N-hexane (99%), sodium carbonate, potassium chloride and Folin-Ciocalteau reagent were supplied by Carlo Erba Reagents (Val de Reuil, France). Sodium hydroxide, hydrochloric acid, sodium chlorite, acetic acid glacial, potassium hydroxide, sulfuric acid that were used for extracting the lignin fraction from egagropili were purchased from Sigma Chemical (St Louis, MO, USA), as well as aluminum chloride, quercetin, glycerol and DPPH. The sodium nitrate used in high-performance size exclusion chromatography SEC-TDA analyses and the sodium acetate used in high-performance anion-exchange chromatography with pulsed amperometric detection (HPAE-PAD) analyses were from Sigma-Aldrich (St. Louis, MO, USA), as well as all the monosaccharide standards used to identify the peaks in HPAE-PAD analyses. The NaOH solution used to prepare the HPAE-PAD buffers was from J.T. Baker (Rijstenborgherweg, Netherlands), while the hydrochloric acid used for the sample hydrolysis was from Carlo Erba (Milano, Italy).

### 3.2. Egagropili Lignin Containing Fraction Preparation

LF was extracted from egagropili by sodium chlorite oxidation technique as previously described [37]. In particular, egagropili balls were first reshaped to rhizome fibers, washed and rinsed vigorously in distilled water and finally dried in an oven at 80 °C for 24 h. The dried egagropili fibers (10 g) were then grinded in a rotary mill (Grindomix GM200, Retsch GmbH, Haan, Germany) at a speed of 1000 rpm for 3 min to a 60-mesh sieve size, dewaxed by means of Soxhlet apparatus, with 200 mL toluene/ethanol (2:1 *v*/*v*) during 24 h and oven-dried overnight at 105 °C. The dewaxed egagropili powder was dispersed at 70 °C for 2 h in 300 mL of 1.7% sodium chlorite solution, brought at pH 3.5 by acetic acid, and the resulting bleached fibers (known as holocellulose) were separated by using filter paper. The obtained filtrate, subjected to extensive dialysis against distilled water, was referred as LF and stored at 4 °C until used. A quantitative analysis, carried out by calculating the dry weight of LF, indicated that 150 mg of LF were obtained from 1 g of grinded egagropili powder. Holocellulose and cellulose fractions were also prepared as previously described [37], and their extraction process was monitored by FT-IR analysis at room temperature by using a FTIR Nicolet 5700 spectrophotometer (Thermo Fisher Scientific, Waltham, MA, USA) equipped with attenuated total reflectance accessory. Infrared spectra analysis was performed using the Omnic software, in the range of 4000–500 cm^−1^ with a spectral resolution of 2 cm^−1^ and by 64 average scan. Light-barrier properties of LF were analyzed by measuring sample absorbance at different wavelengths ranging from 200 nm to 800 nm by using a Spectrophotometer SmartSpec 3000 Bio-Rad (Segrate, Milan, Italy). Finally, a 20 mg sample of the fraction was dissolved in 600 μL of D_2_O at pH 12 and the ^1^H NMR spectrum was recorded at 298 K with a Varian Inova 500 spectrometer (HDO δ 4.8 as internal reference).

### 3.3. Molecular Weight Analysis by Size-Exclusion Chromatography with Triple Detector Array

To analyze the average Mw, polydispersity index (Mw/Mn) and intrinsic viscosity (IV) of LF component(s), LF was concentrated five times on a 3-kDa centrifugal filter device (Centricon, Amicon, USA) at 12,000 rpm and 4 °C (Centifuge Z216MK, Hermle Labortechnik, Wehingen, Germany). Analyses were performed by using a high-performance SEC-TDA (Viscotek, Malvern, Italy), equipped with a gel permeation chromatography system (GPCmax VE 2001, Viscotek, Malvern, Italy) and with a triple detector array module that included a refractive index detector (RI), a four-bridge viscosimeter (VIS), and a laser detector (LS) made of a right-angle light scattering (RALS) and a low-angle light scattering (LALS) detector. Runs were performed by injecting, as previously described [63], 0.1 mL of the LF sample onto two gel-permeation columns (TSK-GEL GMPWXL, 7.8 × 30.0 cm, Tosoh Bioscience, Italy), put in series and equipped with a guard column (TSK-GEL GMPWXL, 6.0 × 4.0 cm, Tosoh Bioscience, Italy) and, by eluting in isocratic conditions at a flow rate of 0.6 mL·min^−1^ with 0.1 M NaNO_3_, pH 7.0, at 40 °C for 50 min. Data were acquired and analysed by using a OmniSEC software program (Viscotek, Malvern, Italy). The instrument was calibrated by using a polyethylene oxide (PEO) standard (22 kDa PolyCAL, Viscotek, Malvern, Italy) [60]. The values of the average Mw of the LF component(s), as well as of both LF polidispersity index (Mw/Mn) and IV, were determined on the basis of all the detector signals by applying the equations reported by the manufacturer (data from Viscotek) and on the basis of the lignin dn∙dc^−1^ value (0.1875) reported in literature (data from Viscotek).

### 3.4. Monosaccharide Determination by High-Performance Anion-Exchange Chromatography with Pulsed Amperometric Detection

The monosaccharide composition of egagropili fibers and of LF, hollocellulose and cellulose fractions was determined by high-performance anion-exchange chromatography with HPAE-PAD by improving a previously reported method [64]. The samples were hydrolyzed by 5 M HCl treatment for 6 h at 100 °C and 600 rpm (Thermomixer comfort, Eppendorf, Germany) and then analyzed by using a high-pressure ion chromatography system (ICS3000; Thermo Fisher Scientific, Italy), equipped with an anion exchange column (Carbopac PA1; Thermo Fisher Scientific, Italy) and a pulsed amperometric detector (reference electrode Ag-AgCl; measuring electrode Au). Runs were performed by eluting in gradient conditions with 1 to 4 mM NaOH and 100 mM NaOH containing 20 mM sodium acetate (0–12 min from 1 to 4 mM NaOH, 12–14 min 4 mM NaOH, 14–16 min from 4 to 100 mM NaOH, 16–30 min 100 mM NaOH + 20 mM NaCH_3_COOH, 30–39 min from 100 mM NaOH + 20 mM NaCH_3_COOH to 1 mM NaOH; 39–41 min 1 mM NaOH), at a flow rate of 1 mL/min for 41 min, by injecting 25 μL of the sample, as previously described [64]. The identity of each monosaccharide peak was determined on the basis of the elution times by comparison with standard solutions of different monosaccharides (Fuc, Arab, Rham, Gal, GlcN, Glc, Xyl, GlcA). Calibration curves of the monosaccharide standards were built in the range from 0.002 to 0.008 g/L (for Fuc, Gal, GlcN, Glc), from 0.02 to 0.08 g/L (for Arab, Rham, Xyl), and from 0.2 to 0.8 g/L (for GlcA) after their acid hydrolysis performed at 2.5 mg/mL as described above. The percentage of representativity of each monosaccharide was calculated according to the following formula:%X = %[X (g/L)/(ΣXn (g/L) × 100](1)
where X is the n monosaccharide, whereas the total carbohydrate content percentage with respect to the dry weight of the samples was calculated according to the following formula:%carbohydrate content = %[ΣXn (g)/(dry weight (g) × 100](2)

### 3.5. Phenol, Flavonoid and Anthocyanin Determination

LF TPC was determined by the Folin–Ciocalteau method as described by Velderrain-Rodríguez et al. [65], with some modifications. An aliquot (0.1 mL) of gallic acid solutions prepared at different concentrations (0.01–0.25 mg/mL) was mixed with 0.75 mL of 2 N Folin–Ciocalteau reagent and 0.65 mL of freshly prepared 7.5% (*w*/*v*) Na_2_CO_3_ solution to obtain a calibration curve for quantifying TPC. Absorbance of the samples was measured after 30 min at 765 nm using UV/visible Spectrophotometer (SmartSpec 3000 Bio-Rad, Segrate, Milan, Italy). Then, 100 μL of LF sample (1.5 mg/mL) were mixed with the same reagents and incubated for 30 min, as performed for the calibration curve, and the absorbance was determined at 765 nm. To measure the polarity of the phenolic content, 100 μL of LF were mixed with 200 μL of n-hexane, vortexed gently and then centrifuged. Then, 100 μL of each obtained phase were mixed with the reagent in the same conditions mentioned for LF sample and the absorbance was recorded. Results were expressed as mg of gallic acid equivalents/g of dried LF.

Flavonoid content determination was performed by the aluminum chloride colorimetric method [66]. Two solutions of 3.5 mg/mL of NaNO_2_ (solution A) and of 18.18 mg/mL of NaOH (solution B), respectively, were preliminarily prepared and, then, 0.1 mL of either LF (1.5 mg/mL) sample or quercetin solution at different concentrations (0.01–2.5 mg/mL) were mixed with 0.43 mL of solution A. After incubation for 5 min, 30 μL of 10% anhydrous AlCl_3_ solution were added and, at the end of a further 1 min incubation, also 0.44 mL of solution B was mixed with the samples. Finally, the absorbance was measured at 415 nm using UV/visible Spectrophotometer (SmartSpec 3000 Bio-Rad, Segrate, Milan, Italy) and the total flavonoid content, expressed as mg of quercetin/g of dried LF, was determined from the obtained quercetin calibration curve.

LF anthocyanin content was determined by the pH-differential method [67]. Then, 25 mM of potassium chloride/0.4 M sodium acetate buffer were prepared at both pH 1.0 and 4.5, respectively. LF was mixed with these two different buffers at a 3:1 buffer/LF ratio and the absorbance of the samples measured at 515 and 700 nm by using distilled water as blank. The anthocyanin content (mg cyanidin-3-gucoside/g of extract) was calculated on the basis of the following equation:Anthocyanin content = [(A_515_ − A_700_) pH 1.0 − (A_515_ − A_700_) pH 4.5] × Mw × DF × 1000/ε × L(3)
where A_515_ and A_700_ are the absorbances recorded at 515 nm and 700 nm, respectively, Mw is the molecular weight of cyanidin-3-glucoside (449.2 g/mol), DF is the dilution factor, L is the cell path length (cm), and ε is the molar extinction coefficient for cyanidin-3-glucoside (26,900 L mol^−1^ cm^−1^).

### 3.6. Antioxidant Activity Analysis

LF activity to scavenge DPPH was carried out according to Parveen et al. [68], with some modifications. Variously diluted solutions of LF (0.1 mL), containing different amounts of LF components (0.03–0.15 mg), were mixed with 0.9 mL of methanol DPPH solution (0.005%, *w*/*v*) and then incubated in dark at room temperature for different times (5–90 min). In addition, the antioxidant activity of LF (0.15 mg) was investigated by adding 0.1 mL of the sample to 0.9 mL of freshly prepared DPPH solution and incubating the aliquots in dark for 90 min during 6 months. Absorbance was measured at 517 nm using UV/visible Spectrophotometer (SmartSpec 3000 Bio-Rad, Segrate, Milan, Italy). Methanol was used as blank, while water (100 µL) added to DPPH solution (900 µL) was used as the control sample. DPPH radical scavenging activity was calculated by the following equation:DPPH scavenging activity (%) = (A_0_ − As)/A_0_ × 100(4)
where A_0_ is the absorbance of the control sample and As the absorbance of the sample. HP-based films, derived from FFSs containing 400 mg of HPs and 50% glycerol as a plasticizer, were prepared in the absence and presence of 6% (*w*/*w* with respect to HP) LF, as previously described [37], and the antioxidant activity of both FFSs and films was evaluated. The films (20 mg) were dissolved in 1.0 mL methanol, whereas the respective FFSs (0.01 mL) were mixed with 1 mL of methanol. Then, 0.1 mL of film and FFS samples, both containing 0.12 mg of LF, were mixed with 0.9 mL of DPPH solution (0.005%, *w*/*v*). The absorbance of each sample was measured at 517 nm after incubation in darkness for 30, 60 and 90 min at room temperature according to Equation (2).

Additional experiments were carried out by immersing pieces of each film (100 mg) in 5 mL of methanol DPPH solution (0.005%, *w*/*v*) and the samples were kept in a dark room without any shaking in order to investigate the profile of LF release in the DPPH solution. Aliquots (0.1 mL) of the solutions were taken at different times, from 0.5 to 24 h, and mixed with 0.9 mL of DPPH solution. The absorbance of the collected samples was finally measured at 517 nm. All the experiments were carried out in triplicate.

### 3.7. Statistical Analysis

SPSS19 (Version 19, SPSS Inc., Chicago, IL, USA) software was used for all statistical analyses. One-way analysis of variance (ANOVA) and Duncan’s multiple range tests (*p* < 0.05) were used to determine the significant difference among the samples. All treatments were analyzed in triplicate and in a completely randomized design.

## 4. Conclusions

A water soluble lignin (27.5%)/carbohydrate (72.5%) complex (LCC) of 34.75 kDa Mw, formed by a phenol polymeric chain covalently bound to hemicellulose fragments, was extracted from *Posidonia oceanica* sea balls (egagropili) and characterized. Glucuronic acid (46.6%) and rhamnose (29.6%) were the most present monosaccharides in the isolated LCC, whereas 22% of the LCC total phenol content was constituted by flavonoids, 10% of which were anthocyanins. LCC exhibited a remarkable and stable antioxidant activity that is possible to incorporate into protein-based films from which it is easily released over time. Egagropili LCC is proposed as an effective antioxidant additive for the reinforcement and the improvement of gas barrier properties of biodegradable materials potentially useful for the packaging of perishable foods sensitive to oxidation.

## Figures and Tables

**Figure 1 ijms-22-09147-f001:**
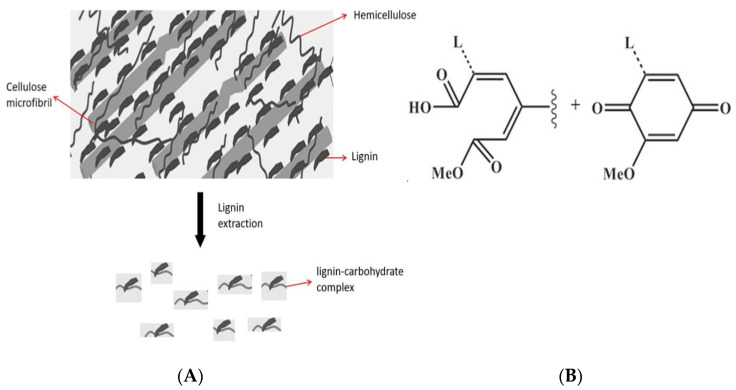
Lignin-carbohydrate complex formation during lignin extraction from a lignocellulosic biomass (**A**). Lignin main fragments obtained from lignocellulosic biomass following NaClO_2_ treatment (Li et al., 2017) (**B**).

**Figure 2 ijms-22-09147-f002:**
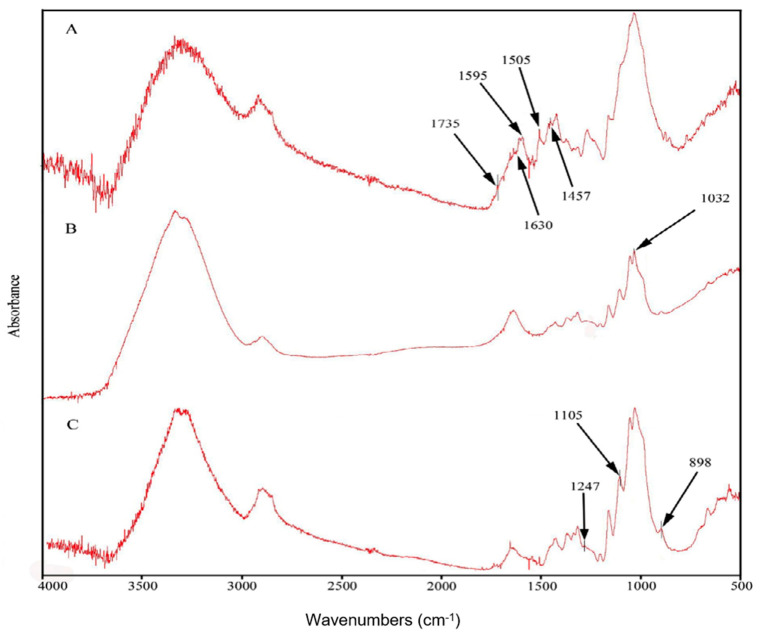
Fourier-transform infrared spectra of fibers (**A**), holocellulose (**B**) and cellulose (**C**) extracted from egagropili dewaxed powder. Further experimental details are given in the text.

**Figure 3 ijms-22-09147-f003:**
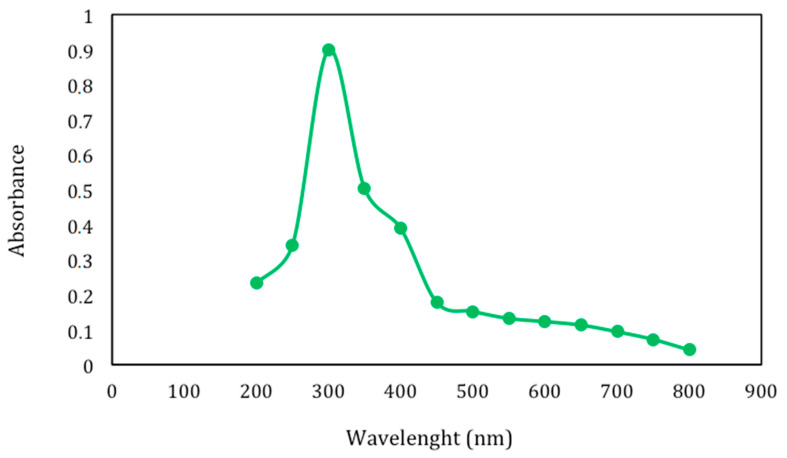
Absorbance at wavelengths between 200 and 800 nm of extensively dialyzed lignin fraction (0.15 mg) extracted from dewaxed egagropili powder.

**Figure 4 ijms-22-09147-f004:**
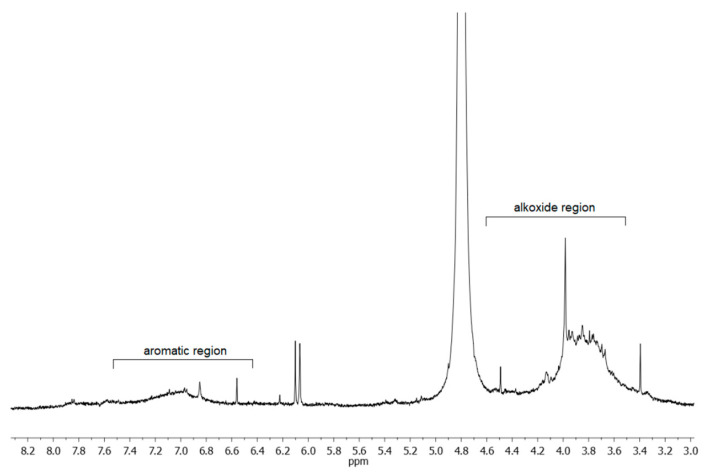
Portion of ^1^H NMR spectrum of the egagropili lignin fraction in D_2_O (pH 12) at 298 K and 500 MHz. The intense peak at δ 4.8 ppm is due to non-deuterated water taken as reference.

**Figure 5 ijms-22-09147-f005:**
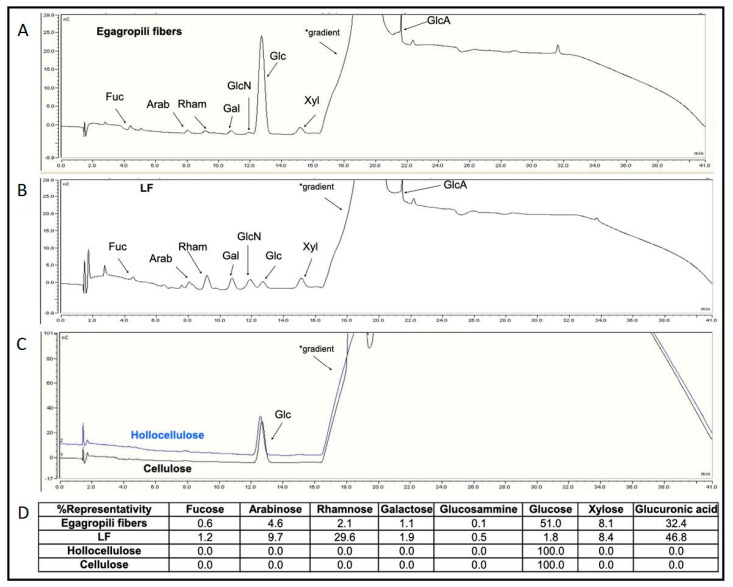
High-performance anion-exchange chromatograms of egagropili fibers (**A**), lignin fraction (LF) (**B**), hollocellulose and cellulose samples (**C**). The different monosaccharide peaks are indicated by the arrows. The representativity percentage of each monosaccharide is reported in the panel (**D**). * Increasing concentration of NaOH used as eluent. Further experimental details are given in the text.

**Figure 6 ijms-22-09147-f006:**
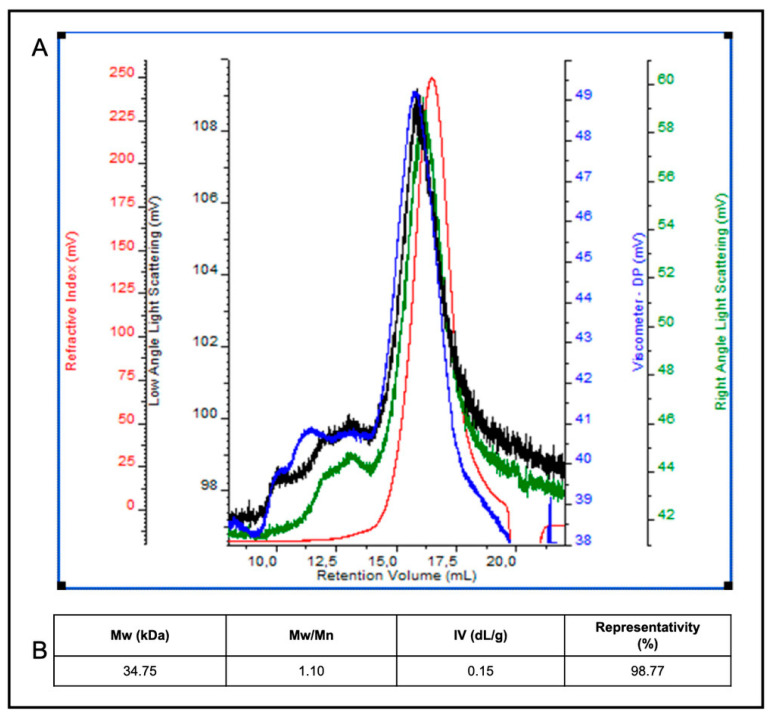
SEC-TDA analysis of egagropili lignin fraction (LF). Panel A shows the chromatogram of overlaid signals of the refractive index (red line), the right and low angle laser scattering (green and black line) and the viscometer (blue line) detectors (**A**). LF component averaged molecular weight (Mw), polydispersity index (Mw/Mn), intrinsic viscosity (IV) and peak representativity values are reported in the panel (**B**). Further experimental details are given in the text.

**Figure 7 ijms-22-09147-f007:**
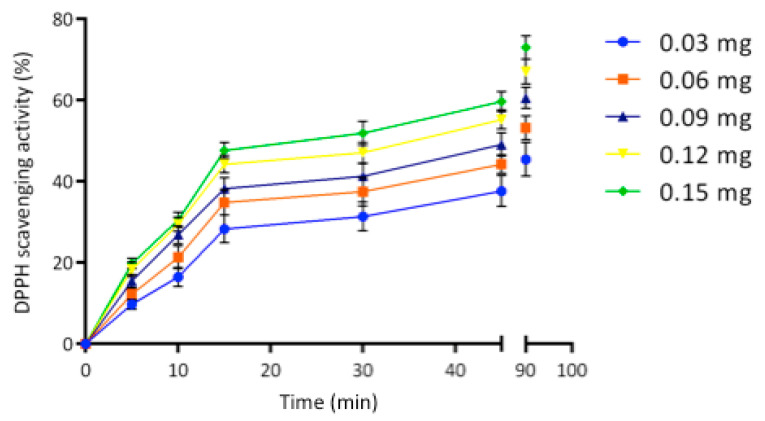
2,2-diphenyl-1-picrylhydrazyl (DPPH) scavenging activity of different concentrations of egagropili lignin fraction during 90 min of incubation of samples at room temperature and dark. Further experimental details are given in the text.

**Figure 8 ijms-22-09147-f008:**
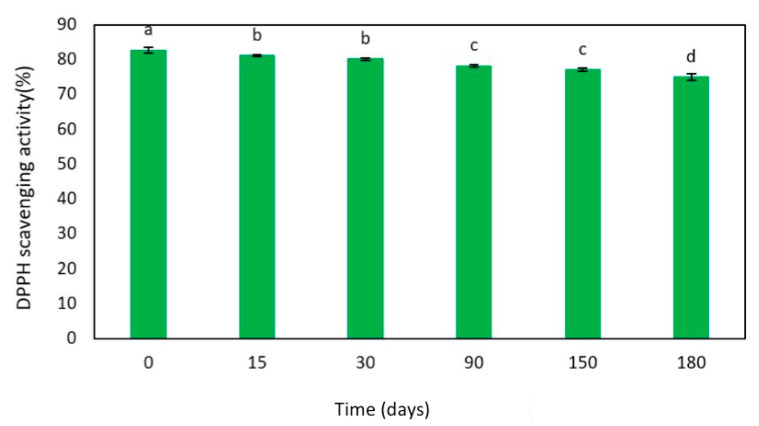
Effect of 6-month time storage of egagropili lignin fraction (0.15 mg) on 2,2-diphenyl-1-picrylhydrazyl (DPPH) scavenging activity during 90 min of sample incubation. Values with different small letters (a–d) were significantly different (*p* < 0.05). Further experimental details are given in the text.

**Figure 9 ijms-22-09147-f009:**
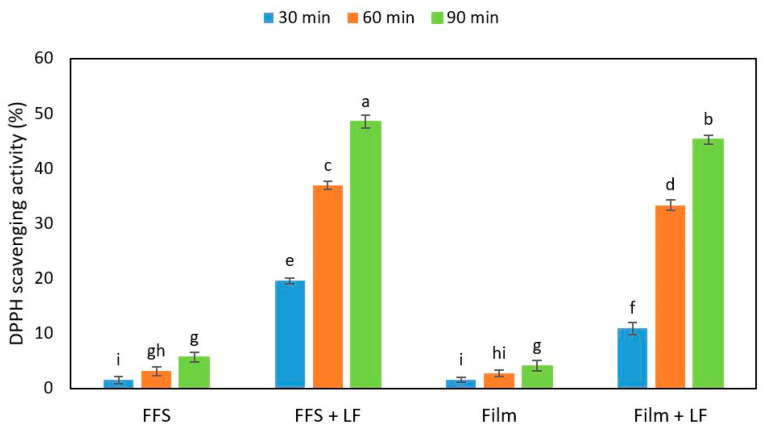
Antioxidant activity of hemp-protein-based film-forming solutions (FFSs) and films prepared in the absence or presence and of egagropili lignin fraction (LF) measured by 2,2-diphenyl-1-picrylhydrazyl (DPPH) assay at different times of incubation. Values with different small letters (a–i) were significantly different (*p* < 0.05). Further experimental details were given in the text.

**Figure 10 ijms-22-09147-f010:**
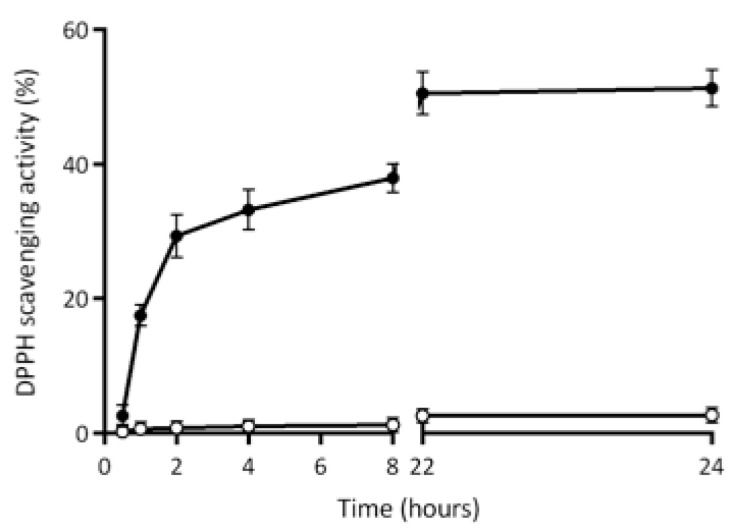
Profile of antioxidant agent release during 24 h in the 2,2-diphenyl-1-picrylhydrazyl (DPPH) solution from the matrix of hemp-protein-based films prepared in the absence (-○-) or presence (-●-) of egagropili lignin fraction (6% *w*/*w* protein). Further experimental details were given in the text.

**Table 1 ijms-22-09147-t001:** Phenol, flavonoid and anthocyanin content of lignin fraction extracted from egagropili.

Total Phenols (*mg gallic acid eqs./g extract*)	113.85 ± 5.87
Hexan-extracted Phenols (*mg gallic acid eqs./g extract*)	1.19 ± 0.63
Flavonoids (*mg quercetin eqs./g extract*)	24.5 ± 0.32
Anthocyanins (*mg cyanidin-3-glucoside/g extract*)	2.60 ± 0.51

## Data Availability

The data presented in this study are available on request from the corresponding author.

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
