# Peer review of "Lignin/Carbohydrate Complex Isolated from Posidonia oceanica Sea Balls (Egagropili): Characterization and Antioxidant Reinforcement of Protein-Based Films"

_ijms, 2021, doi:10.3390/ijms22179147_

Round 1

Reviewer 1 Report

In this study, the authors extract and characterize a lignin-carbohydrate complex from a product of the sea grass Posidonia oceanica. This biomass feedstock is described by the authors to be a prevalent waste/debris occurring  naturally within the environment. After extraction, the authors analyze the LCC fraction with FTIR, UV absorbance, H-NMR, SEC for molecular weight, and various antioxidant quantities/activities. Based on the results of their analyses, the authors conclude that the LCC extracted from this plant matter may be feasible for applications requiring antioxidant additives.

It would be interesting (if possible) to see how this work compares to other recent work on lignin analysis from this sea grass (specifically: Rencoret et al. Deciphering the unique structure and acylation pattern of Posidonia oceanica lignin. ACS Sustainable Chem Eng (2020), 8(33), 12521-12533.)

I would suggest for the authors to please define all acronyms and abbreviations in the text before using the abbreviation. For example, some of the abbreviations found on page 6 may be unclear.

An additional remark is that this may be an issue with my computer/monitor; however, some of the figures are appearing with low resolution (particularly figure 7). Please verify that the figures are of sufficient quality for the journal.

Finally: the materials and methods section is very thorough. I would encourage the authors to review this section to ensure that the maximum amount of detail is provided with respect to reproducibility, as there is a fair amount of analysis and preparatory work that went into this study.

Author Response

Reviewer #1

Comment to the Authors: It would be interesting (if possible) to see how this work compares to other recent work on lignin analysis from this sea grass (specifically: Rencoret et al. Deciphering the unique structure and acylation pattern of Posidonia oceanica lignin. ACS Sustainable Chem Eng (2020), 8(33), 12521-12533.).

Reply of Authors: A comparison with the recent work by Rencoret et al. has been added in the

paragraph 2.2 of the “Results and Discussion section” (p. 8, lines 258-265). As a consequence, the

reference list has been updated with the new reference (ref n. 60).

Comment to the Authors: I would suggest for the authors to please define all acronyms and abbreviations in the text before using the abbreviation. For example, some of the abbreviations found on page 6 may be unclear.

Reply of Authors: All the acronyms and abbreviations have been checked and defined in the text (p. 6, line 192). For the sake of clarity, we have added an “abbreviation list” at the end of manuscript (p. 16).

Comment to the Authors: An additional remark is that this may be an issue with my computer/monitor; however, some of the figures are appearing with low resolution (particularly figure 7). Please verify that the figures are of sufficient quality for the journal.

Reply of Authors: All the figures have been improved verifying that they meet the journal standards.

Comment to the Authors: the materials and methods section is very thorough. I would encourage the authors to review this section to ensure that the maximum amount of detail is provided with respect to reproducibility, as there is a fair amount of analysis and preparatory work that went into this study.

Reply of Authors: The “Materials and methods section” has been reviewed by adding further details (p. 12, lines 349-353; p. 13, lines 393-394, 399-402, 419-425; p. 14, lines 470-471; p. 15, lines 506-510).

Reviewer 2 Report

The authors have isolated a lignin/carbohydrate complex from Posidonia oceanica sea balls (egagropili), characterized and evaluated antioxidant activity for reinforcement of protein-based films. This is an interesting piece of work and technically sound. However, the following points need to be addressed before this paper could be accepted for publication:

  1. The introduction should be updated by citing the following related articles: Int. J. Bio. Macromol. 2009, 45, 146-151; Int. J. Bio. Macromol. 2010, 47, 445-453; Int. J. Bio. Macromol. 2020, 161, 1484-1495.
  2. Figures 1 & 2 can be combined as parts of a same figure.
  3. Figure 5 – NMR peak values and assignments should be provided inside the figure itself.
  4. Table 1 – the column and row should be rearranged for better understanding of its contents.
  5. Figures 6 & 7 – X and Y axis labels and numbers should be provided with clarity and increased size.
  6. All the equations and units should be double-checked for correctness, especially all the variables involved in the equations should be described clearly.
  7. A new sub-section “3.7. Statistical Analysis” should be added containing the information on number of replicates, method used for analysis of significance and statistical software used to perform them.
  8. Both typographical and grammatical errors should be double-checked throughout the manuscript.

Author Response

Reviewer #2

Comment to the Authors: The introduction should be updated by citing the following related articles: Int. J. Bio. Macromol. 2009, 45, 146-151; Int. J. Bio. Macromol. 2010, 47, 445-453; Int. J. Bio. Macromol. 2020, 161, 1484-1495.

Reply of Authors: The introduction has been updated by adding the suggested articles (p. 2, line 41). As a consequence the reference list has been updated with the new references (refs. n. 4-6)

Comment to the Authors: Figures 1 & 2 can be combined as parts of a same figure.

Reply of Authors: The Figures 1 &2 have been combined as a part of the same figure (Please see the new Figure 1) and, as a consequence, the numbers for all figures has changed.

Comment to the Authors: Figure 5 – NMR peak values and assignments should be provided inside the figure itself.

Reply of Authors: NMR peak values and assignments have been provided in the Figure 5.

Comment to the Authors: Table 1 – the column and row should be rearranged for better understanding of its contents.

Reply of Authors: The columns and rows of Table 1 have been rearranged (p. 8).

Comment to the Authors: Figures 6 & 7 – X and Y axis labels and numbers should be provided with clarity and increased size.

Reply of Authors: Figures 6 & 7 have been improved (p. 9 and 10).

Comment to the Authors: All the equations and units should be double-checked for correctness, especially all the variables involved in the equations should be described clearly.

Reply of Authors: All the equations and units have been double-checked and all the variable all the variables involved in the equations described (p. 14; lines 470-471)

Comment to the Authors: A new sub-section “3.7. Statistical Analysis” should be added containing the information on number of replicates, method used for analysis of significance and statistical software used to perform them.

Reply of Authors: A new sub-section “3.7. Statistical Analysis” was added in Materials and Methods section. (p. 15).

Comment to the Authors: Both typographical and grammatical errors should be double-checked throughout the manuscript.

Reply of Authors: Both typographical and grammatical errors were double-checked throughout the manuscript.